# The Mechanical Properties of Blended Fibrinogen:Polycaprolactone (PCL) Nanofibers

**DOI:** 10.3390/nano13081359

**Published:** 2023-04-13

**Authors:** Nouf Alharbi, Annelise Brigham, Martin Guthold

**Affiliations:** Department of Physics, Wake Forest University, Winston-Salem, NC 27109, USA; alhana17@wfu.edu (N.A.);

**Keywords:** electrospinning, fibrinogen, polycaprolactone, mechanical properties, nanofibers, diameter dependence

## Abstract

Electrospinning is a process to produce versatile nanoscale fibers. In this process, synthetic and natural polymers can be combined to produce novel, blended materials with a range of physical, chemical, and biological properties. We electrospun biocompatible, blended fibrinogen:polycaprolactone (PCL) nanofibers with diameters ranging from 40 nm to 600 nm, at 25:75 and 75:25 blend ratios and determined their mechanical properties using a combined atomic force/optical microscopy technique. Fiber extensibility (breaking strain), elastic limit, and stress relaxation times depended on blend ratios but not fiber diameter. As the fibrinogen:PCL ratio increased from 25:75 to 75:25, extensibility decreased from 120% to 63% and elastic limit decreased from a range between 18% and 40% to a range between 12% and 27%. Stiffness-related properties, including the Young’s modulus, rupture stress, and the total and relaxed, elastic moduli (Kelvin model), strongly depended on fiber diameter. For diameters less than 150 nm, these stiffness-related quantities varied approximately as *D*^−2^; above 300 nm the diameter dependence leveled off. 50 nm fibers were five–ten times stiffer than 300 nm fibers. These findings indicate that fiber diameter, in addition to fiber material, critically affects nanofiber properties. Drawing on previously published data, a summary of the mechanical properties for fibrinogen:PCL nanofibers with ratios of 100:0, 75:25, 50:50, 25:75 and 0:100 is provided.

## 1. Introduction

Electrospun nanofibers have gained prominence in recent years due to their versatility and unique properties. Large surface area to volume ratios, nanoscale size, and a wide range of physical and biochemical properties make electrospun fibers an attractive material for various fields such as tissue engineering [1,2,3], medication delivery [4], textile manufacture [5,6], filtration [7,8], and clean energy (batteries, solar panels, fuel cells) [9,10]. Although several techniques exist to generate ultra-thin fibers, electrospinning is one of the most economical and straightforward processes. Electrospinning offers several advantages, including ease of use, scalability, and adjustability [11]. This technique utilizes a high electric field to produce fibers on the nanoscale using polymer solutions of synthetic or natural polymers [12,13]. It allows control of fiber diameter, mesh pore size, and surface morphology [14,15], and, if desired, the fibers may be infused with additional small molecules. Furthermore, electrospinning enables the creation of diverse structures, including hollow [16], core-shell [17], multilayer [18], and nanowires [19], providing great versatility in the nanofiber design for various demands in the applications [20].

Several factors affect electrospinning, including solution composition, processing parameters (flow rate, electric field strength), and ambient conditions (temperature, humidity) [21,22,23,24,25,26]. Understanding and adjusting these parameters allows the production of nanofibers that meet the requirements of specific applications.

Over the past years, natural polymers such as collagen, fibrinogen, and elastin were successfully electrospun to nanofibers for potential uses such as tissue engineering scaffolds, wound dressings, and various other biomedical applications [27,28,29,30,31]. A growing interest exists in creating new materials by blending natural polymers with synthetic ones [32,33]. Blending polymers can produce new materials with distinctive structural, mechanical and biochemical properties. Studies showed that blending natural and synthetic polymers can improve mechanical stability, as natural materials are often weaker than synthetic ones [34,35,36]. Blending synthetic and natural polymers may result in the formation of either covalent or physical bonds between the polymer chains [37,38]. These interlinks may create a strong and stable network structure for some polymers that can significantly enhance the mechanical properties of the resulting material. However, there may also be compatibility problems for other polymers resulting in weaker materials. Further, blending synthetic materials with bioactive proteins may endow nanofibers with biological functionality [39,40,41].

Fibrinogen is a soluble protein with a molecular weight of 340 kDa and is primarily found in the blood plasma. Fibrinogen’s principal function is forming fibrin fibers, which provide a mechanical and structural scaffold for blood clots at the site of an injury to a blood vessel [42]. Electrospun fibrinogen nanofiber-based scaffolds have been successfully fabricated for potential tissue engineering applications [28,43]. Even though pure fibrinogen fibers have excellent properties regarding interactions with cells, such as supporting cell adhesion, proliferation, and differentiation, some applications may require altered mechanical properties [44].

Although fibrinogen possesses appealing characteristics such as high biocompatibility, biodegradability, and the capacity to mimic the natural extracellular matrix, its use in biomedical applications has received less attention compared to other natural polymers like collagen, elastin, and chitosan. Moreover, only a few studies have investigated the potential for blending fibrinogen with synthetic polymers. Hence, further research is needed to fully understand the potential of fibrinogen as a biomaterial and optimize its properties for specific applications.

Polycaprolactone (PCL) is a synthetic polyester polymer with a partially crystalline structure and a low melting point (~60 °C). PCL has been widely used in biomedical applications due to its biodegradability, biocompatibility, mechanical stability, and relative softness [45,46]. Moreover, electrospun PCL networks can mimic the structure of the native extracellular matrices (ECM) [47,48]. Hence, blending fibrinogen with PCL may produce a new biomaterial that meets mechanical and biochemical scaffold design requirements.

Polymeric materials must fulfill several requirements to be suitable as a scaffold for biomedical applications. Among these requirements, mechanical properties are essential when fabricating scaffolds. The ideal scaffold should mimic the physical and chemical structure of the microenvironment of the ECM. Moreover, it should possess enough mechanical strength and integrity to withstand various forces when handled or implanted. For example, electrospun scaffolds for tissue engineering vascular grafts may serve as a conduit for blood flow. Thus, it must sustain the forces put on it without rupturing or being permanently deformed as in an aneurysm [49].

Until recently, little research was done in determining the mechanical properties of single electrospun nanofibers. Previous research mostly investigated the mechanical properties of the entire scaffold or mat. The mechanical properties of scaffolds strongly depend on several factors such as network porosity, single fiber properties, fiber junctions, fiber distribution, and the alignment of the fibers [50,51,52]. Obtaining precise information on how various factors affect scaffold mechanical strength is challenging. Prior knowledge of the modulus and strength of the single fibers forming the mat is critical to build a mechanical model of scaffolds. 

The atomic force microscope (AFM) is one of the most suitable tools for the mechanical analysis and topographic characterization of soft matter, such as polymers, biological materials (proteins, DNA, cells), colloids, nanoparticles, and soft materials, such as gels, elastomers, and hydrogels [53]. A microindentation test can be conducted using AFM, which involves applying a controlled load to a small probe tip and measuring the resulting indentation depth on the sample’s surface. The force–indentation depth curve obtained from microindentation testing can provide information about the material’s mechanical properties, such as the elastic modulus, hardness, and toughness [54]. 

In addition, the atomic force microscope (AFM) can perform tensile tests on soft materials by connecting the AFM tip to the sample and then stretching it until it breaks while simultaneously measuring the force and displacement. The resulting force–displacement curve obtained from tensile testing can provide valuable data about the material’s tensile strength, strain at break, and elasticity under tension.

We used a combined AFM and optical microscope to investigate the mechanical properties of electrospun, blended 75:25 and 25:75 fibrinogen:PCL nanofibers. The current work expands the results of previous studies and in total, we have now investigated the mechanical properties of electrospun fibrinogen:PCL nanofibers with ratios of 0:100, 25:75, 50:50, 75:25 and 100:0 [55,56,57]. Taken together, this work provides a library of mechanical properties for dry, electrospun fibrinogen:PCL nanofibers. 

## 2. Materials and Methods

### 2.1. Materials 

Polycaprolactone (average molecular weight, *MW* = 80,000 g/mol), bovine fibrinogen powder (>75% clottable, final concentration 100 mg/mL), and Hexafluoro-2-propanol (HFP) were purchased from Sigma-Aldrich (St. Louis, MO, USA). Dulbecco’s Modified Eagle’s Medium was obtained from Thermo Fisher Scientific (Waltham, MA, USA). Blunt needles were purchased from CML supply (Lexington, KY, USA). A PHD 2000 Infusion syringe pump was obtained from Harvard Apparatus (Holliston, MA, USA). Marine Loctite Epoxy was purchased from Henkel Loctite Corporation (Rocky Hill, CT, USA)

### 2.2. Methods 

#### 2.2.1. Preparation of Solutions with Two Different Ratios

PCL and fibrinogen solutions were prepared separately with a concentration of 100 mg/mL by dissolving PCL pellets in HFP and the fibrinogen powder in 9 parts of HFP and 1 part of Dulbecco’s Modified Eagle’s Medium solvent. The two solutions were stirred with a magnetic stirrer at room temperature for 4 h. The solutions appeared homogeneous. The solutions can be kept for several days when stored in a sealed container at room temperature. To create a solution with two different ratios, 7.5 mL of fibrinogen solution was added to 2.5 mL of PCL solution to make a 75:25 fibrinogen:PCL mixture, and 2.5 mL of the fibrinogen solution was mixed with 7.5 mL of PCL solution to make a 25:75 fibrinogen:PCL mixture. The solution was then stirred for several hours until it became homogenous.

#### 2.2.2. Fabrication of Aligned Nanofibers

About 1 mL of the fibrinogen:PCL solution was transferred into a 5 mL syringe attached to a 20 gauge, 1 inch long, blunt syringe needle. The needle was attached to about 15 cm of silastic medical-grade tubing that had a blunt needle also at the other end. The syringe was placed in a syringe pump, and the solution was released at a constant rate of 0.8 mL/h. The needle at the end of the silastic tubing was connected to 20,000 V via an alligator clip and placed 19 cm from the collection site. We selected these parameters after conducting several preliminary experiments to optimize parameters. The collection site consisted of two grounded copper plates that are 2 cm apart from each other; the entire collection site is connected to ground. The glass cover slide, to which the striated substrate is affixed, is taped via conducting copper tape to both the copper plates (Figure 1A). The two strips of copper tape on the top and bottom of the cover slide are utilized to collect aligned nanofibers, as reported in [58]. The stream of the solution is alternately attracted to both strips of copper tapes, resulting in fiber alignment across the ridges (Figure 1B). A schematic of the set-up is displayed in Figure 1A. The fibers were prepared in a large, homemade plexiglass box at lab ambient conditions (23 °C and about 35% humidity).

#### 2.2.3. Anchoring of Nanofibers to the Ridges

As was observed in previous experiments conducted in this laboratory [59], electrospun PCL fibers slip on the ridges when pulled with the atomic force microscopy (AFM) probe. Thus, fibers containing PCL needed to be anchored to the ridges of the substrate. To do this, 10 µL of Marine Loctite Epoxy were deposited with a pipette onto the cover slide, next to the striated substrate. We used a piece of a cover slide to carefully spread the epoxy on the substrate to thin it out. This ensured that the AFM probe would be less likely to break or get stuck in the glue in the following dip-pen nanolithography [60] step. Using the Asylum AFM software, a used (somewhat blunted) AFM tip was gradually lowered into the epoxy and raised quickly to avoid excess epoxy. Older, blunted AFM tips worked better at picking up epoxy than new, sharp tips. After locating a fiber perpendicular to the ridges, the tip was carefully lowered onto both the ridge and the fiber, and left in contact for several seconds, thereby transferring a small amount of epoxy from the tip to the ridge. Care must be taken not to transfer too much epoxy, as a fiber becomes unsuitable for measurements if the epoxy runs into the groove or travels along the fiber. About 40 fibers can be glued in 90 min. After a desired number of fibers has been glued, the glue was cured for several hours under ambient conditions.

#### 2.2.4. Manipulation of Nanofibers Using AFM

We evaluated the mechanical properties of the nanofibers with a combined AFM/optical microscopy technique. The cover glass with the sample was placed on the microscope stage. The stage serves an inverted optical microscope (Olympus IX 73) and AFM (MFP-3D-Bio, Oxford Instruments-Asylum Research, Santa Barbara, CA, USA). We adjusted the stage before manipulations began, so that the sample was directly above the objective lens and beneath the AFM probe. The AFM was then put into contact mode to begin fiber manipulations. The probe was raised to hover above a specific fiber. The AFM probe was lowered in 500 nm steps until the apex of the tip was horizontally level with the desired fiber. By adjusting the lateral x-y-position, the AFM probe was placed at the middle of the suspended fiber. Only the tip of the AFM probe was in contact with the fiber. The fiber was pulled in the horizontal plane, perpendicular to the fiber length, parallel to the groove, at a speed of 350 nm/s, until it broke (Figure 1C,D). The AFM and the sample were checked regularly to ensure that they were level with each other. Asylum Research AFM Probes AC240TSA-R3 were used (resonance frequency *f* = 70 kHz, spring constant, *k* = 2 N/m, length *L* = 240 µm, width *w* = 40 µm, height *h* = 14 µm, nominal tip radius, *r* = 7 nm). 

#### 2.2.5. Young’s Modulus, Stress, and Strain Measurement

We examined the maximum extensibility, $\varepsilon_{max}$, of the electrospun fibrinogen:PCL nanofibers by simply stretching the fibers parallel to the ridges at rate of 350 nm/s until they ruptured. The initial slope of the stress–strain curve represents the stiffness (Young’s modulus) of the fiber (assuming small, elastic deformations)

The Young’s Modulus, *Y*, of the fiber is defined as
(1)$$Y=\frac{\sigma }{\varepsilon }.$$

σ and ε are the engineering stress and strain, in which the fiber diameter is assumed to be constant during the measurement. The engineering strain, *ε*, and stress *σ* are given as
(2)$$\varepsilon =\frac{L_{f}-L_{i}}{L_{i}}\times 100\%$$
(3)$$\sigma =\frac{F_{fiber}}{A}.$$

$L_{i}$ and $L_{f}$ are the initial and final lengths of the fiber (Figure 1D). *F_fiber_* is force applied force to the fiber and can be determined as previously described [56] and *A* is the cross-sectional area of the fiber, which is assumed to be circular;
(4)$$A=\frac{\pi D^{2}}{4}.$$

*D* is the fiber diameter. 

#### 2.2.6. Energy Loss Calculation

The energy dissipated during a cyclic stress–strain curve (stretching and returning to starting point) is the area between the forward and backward pulls. We analyzed over 55 single curves from each group using OriginLab software (OriginLab Corp., Northampton, MA, USA). For each single curve, the area between and under the curve were integrated and the energy loss defined by dividing the area between loading and unloading curves by the whole area under the loading curve. 

#### 2.2.7. Incremental Stress–Strain Curves

Incremental stress–strain curves are an approach to separate the elastic and viscous components of the mechanical properties of viscoelastic polymers. In this approach, the fiber is strained in small increments (here, ~10% strain), with a pause after each incremental stretch (here, ~1 min). During the pause, the stress relaxes and asymptotically approaches a relaxed stress value. Electrospun fibrinogen:PCL fibers relaxed with a double exponential function and asymptotically approached a constant, non-zero stress value. The appropriate model to fit stress relaxation curves with these characteristics is the Kelvin model depicted in Figure 1E. It consists of a spring with modulus *Y*_0_ in parallel with two dashpot-spring elements with moduli *Y*_1_ and *Y*_2_, and viscosities m_1_ and m_2_, respectively. The equation to fit individual stress relaxation curves is thus,
(5)σ(t)=σ0+σ1×e−tτ1+σ2×e−tτ2
where, *σ*_0_ is the relaxed stress value of the fiber as *t* → ∞. *σ*_1_ and *σ*_2_ are stress prefactors used to determine the total modulus; *Y_tot_ = Y*_0_
*+ Y*_1_
*+ Y*_2_, where *Y_i_ = σ_i_/ε*, with *i* = 0, 1, 2, and *ε* is the strain value at which the fiber is held. *Y*_0_ is the relaxed, elastic modulus. $\tau_{1}$ and $\tau_{2}$ are the fast and slow relaxation time, respectively. The relaxation time, viscosity and modulus of the in-series elements are related by *τ_i_ = η_i_/Y_i_*, with *i* = 1, 2. 

Individual stress relaxation curves were fitted to this double exponential function in Origin (OriginLab Corporation, Northampton, MA, USA). 

#### 2.2.8. Scanning Electron Microscopy 

Scanning electron microscopy (SEM) images of electrospun fibers were taken with a Zeiss Gemini SEM 300 (Carl Zeiss, Oberkochen, Germany). The sample was prepared by cutting the glass slide with the electrospun fibers into a small square shape (1 cm × 1 cm) and mounting it on an aluminum stub using conductive tape. The sample was then sputter-coated with a thin layer of gold. SEM images were collected using an accelerating voltage of 5 kV, a working distance of 7.2 mm, and a magnification of 1140×.

#### 2.2.9. Fiber Diameter Measurement

The AC mode of the AFM was used to determine the diameter of individual fibers. Fibers on the top of the ridge were scanned with a scan rate of 0.4 Hz. The topography images were then processed using Gwyddion software program (http://gwyddion.net/ (accessed on 1 July 2022), Brno, Czech Republic) [61]. 

#### 2.2.10. Statistical Analysis 

All data are presented as the mean ± standard deviation of the mean (mean ± SD). Statistical tests were used to compare the two independent groups. The statistical analysis was carried out using *t*-test (and nonparametric tests). A *p*-value < 0.05 was considered statistically significant and is indicated with * in comparative bar graphs; ** indicates a *p*-value < 0.01; *** indicates a *p*-value < 0.001; **** indicates a *p*-value < 0.0001.

## 3. Results

### 3.1. Fiber Stress and Strain Curves (Maximum Extensibility)

The strain at the breaking point (maximum extensibility) of the fibrinogen:PCL fibers was determined by pulling the fiber with the AFM tip at a rate of 350 nm/s until it broke. Representative stress–strain curves of 75:25 and 25:75 fibrinogen:PCL electrospun fibers are shown in Figure 2A. Both curves show significant strain softening, which means the fibers soften (slope decreases) as the strain increases. 

The 75:25 fibrinogen:PCL fibers exhibited a linear slope until 12% strain (the yield point); then, the slope decreased significantly, and the fiber broke at 69% strain. The Young’s modulus decreased from 351 MPa (initial slope) at a strain of 10% to around 15 MPa (final slope) at 60%.

The fiber became softer by increasing the ratio of the PCL in the mixture from 25% to 75%. This fiber could be stretched to 60% strain before reaching the yield point (Figure 2A). The stress of the plateau region of the stress–strain curve decreases slowly until failure of the fiber at a strain of 125%. The average maximum strains for the 75:25 fibrinogen:PCL and 25:75 fibrinogen:PCL nanofibers were 63 ± 29% and 120 ± 17%, respectively. *t*-test analysis showed a significant difference in the extensibility of the fibers with the two different ratios (*p* < 0.0001) (Figure 2B). The distributions of the extensibilities are shown in Figure 2C,D. Moreover, we determined the maximum stress (breaking stress) as a function of fiber diameter, as shown in Figure 2E,F. The maximum extensibility, maximum stress, Young’s modulus, and other properties of the fibers are listed in Table 1 and Table 2. 

### 3.2. Dependence of Stiffness on Fiber Diameter

To investigate the effect of diameter on the Young’s modulus, *Y*, of the fiber, we evaluated fibers with diameters ranging from 30 nm to 600 nm. The initial, linear portion of the stress–strain curves (Figure 2A) represented the stiffness of the nanofibers. Figure 3A,B show a substantial increase in the Young’s modulus, as the diameter of the fibrinogen:PCL fibers with two different ratios (75:25) and (25:75) decreases. The modulus, *Y*, at the smaller diameters (e.g., 50 nm) was five–ten times larger than the modulus at the larger diameters (e.g., 300 nm). A power law function of the form $Y\left(D\right)=a\cdot D^{-b}+c$ fitted all data well (R^2^ = 0.97–0.99), where *D* was the fiber diameter, *b* was the exponent, and *c* was the modulus of the fibers at large diameters (as *D* → ∞, but practically as *D* > 300 nm). Figure 3A,B show that there was a shift in behavior of the modulus values. At a fiber diameter less than 150 nm, the modulus increased steeply with decreasing diameter. Above diameter values of 150 nm, the modulus slightly decreased to reach a value of 160–270 MPa. The investigated diameter values of the nanofibers ranged from 42 nm to 600 nm for the 75:25 ratio and 49 nm to 517 nm for the 25:75 ratio. The fibers had an average diameter of 188 ± 115 nm and 286 ± 161 nm for the 75:25 and 25:75 fibrinogen:PCL samples, respectively.

### 3.3. Viscoelastic Properties 

#### 3.3.1. Incremental Stress–Strain Curves, Total and Relaxed, Elastic Moduli 

The total and relaxed, elastic moduli of the fibers were separated and analyzed using the incremental stress–strain curves method, in which a fiber is stretched in a series of strain increments. In this experiment, the fiber was pulled to a low strain (typically 10–30%) and held constant for some time (~60 s) at that strain to allow stress relaxation. Then another strain increment was applied and the fiber was held constant for the same time interval. This process was repeated about five times until the fiber broke or permanently deformed. Figure 4A shows a fiber that was stretched to strains of 9%, 18%, 32%, 56%, and 84% and held constant for approximately 30 s. The time vs. stress curve in Figure 4B shows that the fiber gradually relaxed (stress relaxation) at those constant strains. We found that the total and relaxed, elastic moduli exhibited diameter dependence, in which the moduli of fibers with smaller diameters were substantially higher than those of fibers with larger diameters (Table 2). We observed a gradual decrease in the moduli with increasing strains. In Figure 4D, the total and relaxed, elastic moduli at a strain value of 9% were 663 MPa and 511 MPa, whereas, at the strain value of 84%, the moduli were 132 MPa and 102 MPa, respectively.

#### 3.3.2. Stress Relaxation 

The stress, as seen in Figure 4B, decreased over time when a fiber was held at constant strain. Each single stress relaxation curve was fitted with a double exponential decay function to extract a fast and slow relaxation time, as shown in Figure 4C. The average fast and slow relaxation times, *τ*_1_ and *τ*_2_, were found to be 2.4 ± 1.4 s and 14.2 ± 7.5 s for the 75:25 fibrinogen:PCL nanofibers, and 4.7 ± 3.5 s and 21.0 ± 15.9 s for the 25:75 fibrinogen:PCL nanofibers, respectively.

#### 3.3.3. Elasticity 

The elastic limit is the maximum strain a fiber can sustain before being permanently deformed. We determined this property via cyclical stress–strain curves. An individual fiber was pulled to a low strain, followed by returning the AFM tip to its starting position to allow fiber to relax. The pull and return cyclical process repeated at increasing strains until the fiber ruptured or permanently deformed. The fiber was pulled at a rate of 200 nm/s. The curves of the successive pulling cycles of the fibers are shown in Figure 5A,B. The first two curves, red and green, show that the elastic limit had not been reached yet, as the stress for both curves returned to zero as the strain returned to zero. However, the black curve shows the fiber just before the permanent deformation and the blue curve shows that the elastic limit was exceeded as the stress reached zero before the strain, which means this fiber has some slack due to the permanent deformation. We found that the elastic limit, *ε_elastic_*, of the fiber with different volume ratios is between 12 ± 4% and 27 ± 10% (*N* = 32) for 75:25 fibrinogen:PCL and 18 ± 6% and 40 ± 14% (*N* = 33) for 25:75 fibrinogen:PCL.

#### 3.3.4. Energy Loss 

The cyclic stress–strain curves were also used to determine energy loss during the stretching cycle. The area between the loading and unloading stress–strain curves represents the dissipated energy (Figure 5C). Figure 5A,B show that the first cycle (red curves) dissipated a small amount of energy at strains of 2–5% compared to the fourth cycle (blue curves) that dissipated much more energy at strains of 28–33%. The dissipation energy correlated significantly with the strain intervals, as shown in Figure 5D. The percentage of energy loss grew steadily as the strain increased. Moreover, the 75:25 fibrinogen:PCL ratio fibers show a consistently higher energy loss than the 25:75 ratio fibers at the same strain interval. At high strain (>60), the energy loss is 84% and 67% for 75:25 and 25:75 ratios, respectively. 

## 4. Discussion

We employed the electrospinning technique to fabricate blended fibrinogen:PCL fibers with two different ratios (75:25) and (25:75). These hybrid fibers had diameters ranging from about 40 nm to 600 nm. Several mechanical properties, including maximum strain and stress, moduli, energy loss and elasticity of the electrospun fibers, were determined using a nanofiber pulling technique based on a combined AFM/inverted optical microscope. A summary of the total findings is given in Table 1 and Table 2.

As a part of our experimental procedure, we glued the nanofibers to the ridges in the substrate with epoxy to prevent them from slipping when pulled by the AFM probe. Due to the tendency of fibers with high PCL content to slip, as reported before [59], it was necessary to anchor these fibers so that the mechanical properties could be measured accurately. In some cases, the epoxy leaked into the grooves or along the fibers; these fibers were not used for measurements. We allowed the glue to cure at ambient temperature for at least 24 h before manipulation to ensure they were adequately fixed to the ridges.

Drawing on data from previous work [55,56,57], Table 3 summarizes the key mechanical properties of electrospun fibrinogen:PCL fibers with ratios of 100:0, 75:25, 50:50, 25:75, 0:100. This provides a fuller picture of how the different ratios affect nanofiber properties. Pure PCL fibers display the largest extensibility at 133%, which drops slightly and linearly with increasing fibrinogen concentrations to 120% (25:75) and then 110% (50:50). As the fibrinogen:PCL ratio increases to 75:25, the extensibility strongly decreases to 63%, less than half the extensibility of electrospun pure PCL fibers, as seen in Figure 6. This strong decrease is somewhat surprising as the extensibility of pure fibrinogen increases again to 110%. One explanation for this trend in the extensibility could be that PCL dominates blended nanofiber properties for ratios of 50:50 and higher. At these higher PCL concentrations, the PCL polymer chains may still be able to interact with each other and are entangled with each other. For the 75:25 fibrinogen:PCL fibers, the PCL polymers may lose their ability to interact with each other and their entanglement is diminished. The increased extensibility and the high modulus of the pure fibrinogen fibers, indicates that the interactions between fibrinogen polymers in the pure fibrinogen fibers are strong, and likely mediated through long flexible chains. These results also suggest that the fibrinogen polymers and PCL polymers may not interact as well with each other as PCL–PCL interactions and fibrinogen–fibrinogen interactions. 

The effect of the PCL ratio on fiber extensibility we observed agrees with the trend seen in previous reports. Miele et al. found that doubling the weight ratio of PCL in a mixture of PCL and collagen increased strain at the failure from 12.8 ± 0.8% to 63 ± 9% [62]. Mobarakeh showed that the elongation of PCL/gelatin nanofibers significantly increased from ~35% to ~170% when the weight ratio of PCL increased from 50% to 70% [63]. The latter two studies evaluated the mechanical properties of aligned fiber scaffolds rather than single fibers.

Like the extensibility, the elasticity of the blended fibers also increased when the PCL fraction in fibrinogen:PCL fibers was increased. The 25:75 fibrinogen:PCL fibers were permanently deformed between 18 ± 6% and 40 ± 14% strain. In contrast, the deformation of the 75:25 fibrinogen:PCL fibers occurred between 12 ± 4% and 27 ± 10% strain. Sharpe et al. reported a much lower elastic limit for 50:50 fibrinogen/PCL fibers [56]. They found that the 50:50 fibers could be stretched to 5 ± 5% strain before permanent damage was seen. It is not clear why the elastic limit of the 50:50 fibers was much smaller compared to the other ratios, although we followed the same fiber preparations and the same experimental method. However, comparing our findings for blended fibers to pure fibrinogen and pure PCL fibers, the elastic limit of the blended fibers with two different ratios is within the elastic limit range of pure PCL (24%) and pure fibrinogen (15%), as seen in Table 3.

The stiffness (Young’s modulus) of the fibrinogen:PCL fibers strongly depends on the fibrinogen:PCL ratio. At first, we will compare large diameter fibers, ignoring the strong diameter dependence of the small diameter nanofibers. Electrospun, pure fibrinogen fibers are stiffest with a modulus of 4200 MPa, and electrospun pure PCL fibers are softest with a modulus of 380 MPa. The modulus of blended fibers generally decreases with deceasing fibrinogen. The modulus is dominated by the PCL properties for the 0:100 and 25:75 fibrinogen:PCL fibers with both having a modulus around 300 MPa, which is similar to the modulus of bulk PCL [64]. The modulus increases by a factor of about 5 for the 50:50 fibers, and by another factor of about 3 for the pure fibrinogen fibers (100:0). In the 75:25 fibrinogen:PCL fibers there may be a compatibility discrepancy between fibrinogen and PCL, which is why the trend of increasing modulus is broken for these fibers. 

Another key finding is that the Young’s modulus, the rupture stress, and the total and relaxed, elastic moduli (from incremental stress–strain curves) strongly depend on fiber diameter. Fibers with smaller diameters are significantly stiffer than fibers with larger diameters and stiffer than bulk (for PCL), as seen in Table 2. A power law fit to the data yields an exponent of approximately −2 for fibers with a diameter less than about 150 nm; above this diameter, the moduli approach a constant value. This diameter dependence is unusual because the modulus of a homogeneous material is a material constant that should be independent of the dimensions of the material. Intriguingly, this diameter dependence may be a general phenomenon for electrospun and some natural nanofibers. It was recently reported for pure electrospun PCL fibers [57], for which the Young’s modulus strongly increased as the diameter decreased below 100 nm; however, fibers with a diameter greater than 100 nm showed a weak dependence on diameter. It was also seen in natural fibrin fibers [65]. For the electrospun, pure fibrinogen fibers and for the 50:50 fibrinogen:PCL fibers, the diameter dependence was not investigated. 

The diameter dependence may be explained by (at least) two different models. In one model, the nanofiber does not have a uniform cross-sectional density, but instead it has a higher density in the center (core) of the fiber, which decreases toward the periphery. Such a model was invoked by Li et al. to explain the diameter dependence of the modulus for fibrin fibers [65], which form the mechanical and structural scaffold of a blood clot. These authors proposed a fibrin fiber model in which the protofibril density of the fiber decreases with increasing diameter, *D*, as *Y* ∝ $D^{-1.6}$. Similarly, this model was also used by Alharbi et al. to explain the diameter dependence of electrospun PCL fibers [57]. In a second model, the increases in the modulus are due to the higher surface-to-volume ratio of smaller fibers compared to larger fibers, and it assumes that the polymer chains in the surface region of the fiber are much more oriented and aligned than the fibers in the core [66]. Our data cannot conclusively distinguish between these two models, and more studies are needed to validate molecular models. For instance, experiments that could determine the density and alignment of polymers within nanofibers as a function of diameter might be able to shed light on the molecular origins of the observed diameter dependence of the modulus. 

In cyclic stress–strain curves, all blended fibers with different ratios showed a monotonic increase in energy loss with increasing strains. The strong dependence of dissipated energy on strain has been previously reported for several natural and electrospun fibers, such as fibrin fibers, collagen, and electrospun PCL [57,67,68]. For electrospun PCL fiber, the energy loss increased from 38% at small strain (10%) to 66% at high strain (70%). We can estimate the loss energy of electrospun fibrinogen fibers from the elastic limit reported in [55] to be 14% at 3% strain to roughly 65% at 13% strain. Significant energy loss at small strain has also been observed for the electrospun collagen, as the dissipated energy at 12% strain was 80%. Our findings showed that the energy loss increases from 28% at a small strain of <10% to 76–85% at larger than 70% strain for (75:25) and (25:75) fibrinogen:PCL fiber, respectively. The energy loss seems to reach a plateau at a strain of 30–40% with an energy loss of 72–68%. The large energy loss indicates a large viscous component of the deformation.

Besides the fact that these data contribute to building a library that principally focuses on the mechanical properties of single electrospun fiber, the data also provide insight into the mechanical behavior of hybrid fibers, which will help to optimize and design materials suitable for use as scaffolds in many biomedical applications. Future work should focus on understanding how the mechanical properties of individual fibers contribute to the mechanical behavior of meshes made up of these fibers. Additionally, the diameter dependence of the modulus is an intriguing novel property of nanofibers and warrants further exploration. 

## 5. Conclusions

The goal of the present study was to determine the mechanical properties of electrospun fibrinogen:PCL nanofibers as a function of fiber diameter and fibrinogen:PCL ratios. A combined atomic force microscopy/optical microscopy technique was used to determine the mechanical properties of the electrospun hybrid fiber. Extensibility, elasticity, and fast and slow relaxation times depended on the fibrinogen:PCL ratio, as their values increased when the PCL ratio increased from 25% to 75%. Stiffness-related properties, including breaking stress, Young’s, total, and relaxed, elastic moduli strongly depend on the fiber diameter, in addition to the fibrinogen:PCL ratio. Below a fiber diameter of 150 nm, the value of these properties strongly increased, and with decreasing diameter, above 150 nm, the diameter dependence leveled off. These data complement previous data on electrospun, blended fibrinogen:PCL nanofibers, thus completing a library of mechanical properties for fibrinogen:PCL nanofibers with ratios ranging from 100:0 to 0:100. 

The diameter dependence of stiffness-related properties is intriguing and future studies should focus on investigating the molecular mechanisms of this observation. Additionally, the blended fibers should be tested in biological and other applications. 

## Figures and Tables

**Figure 1 nanomaterials-13-01359-f001:**
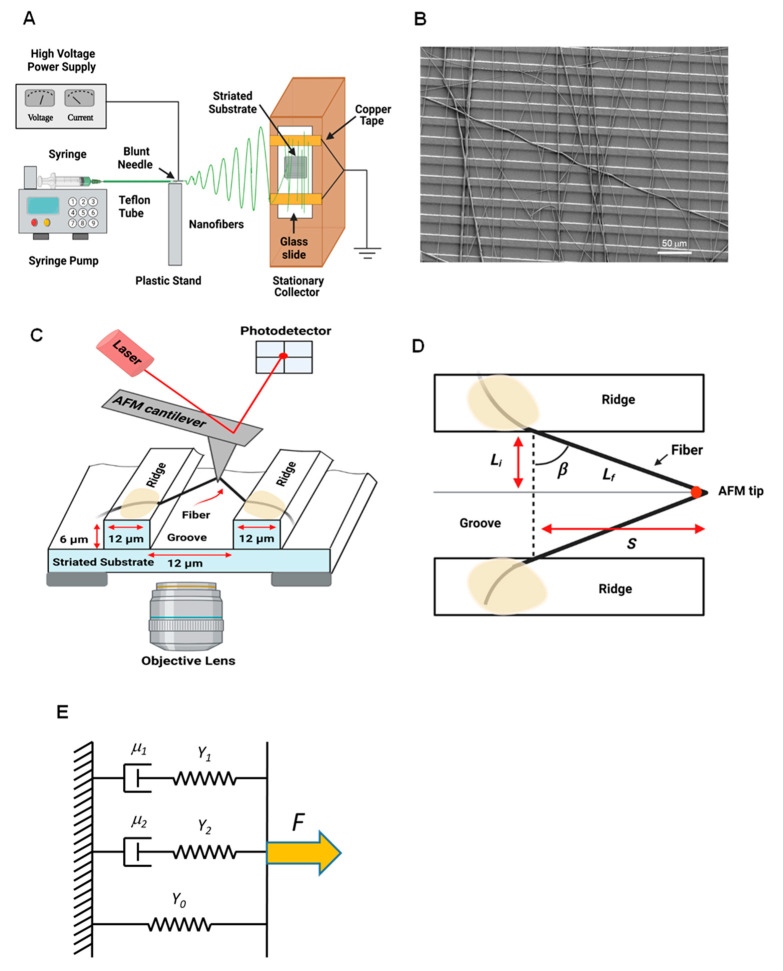
(**A**) Schematic diagram of set up of electrospinning apparatus. (**B**) SEM image of partially aligned, electrospun fibrinogen:PCL fibers on striated substrate. The gap between two ridges is 12 µm. (**C**) Schematic diagram of the AFM manipulating a nanofiber. (**D**) Top view of a pulled fiber. (**E**) Kelvin model of three mechanical elements in parallel. Figure adapted from [56]. Parts of Figure 1 were created with BioRender.com.

**Figure 2 nanomaterials-13-01359-f002:**
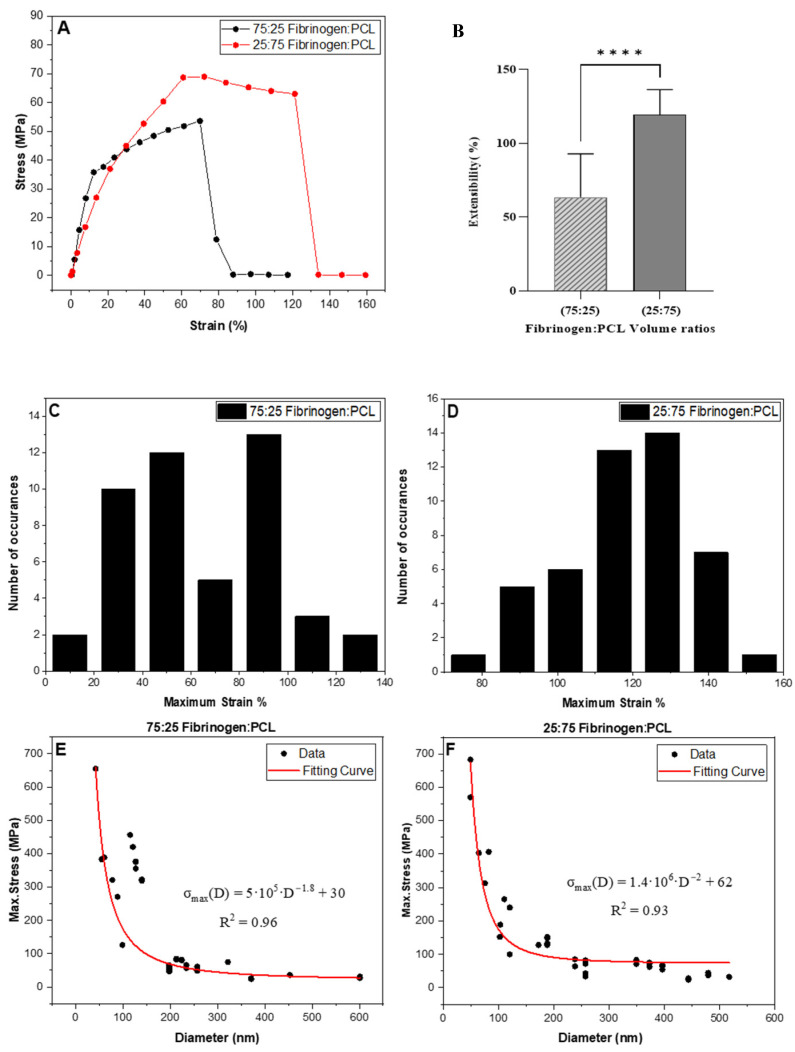
(**A**) Representative stress–strain curves of electrospun fibers showing strain softening. (**B**) Average extensibility of 75:25 and 25:75 fibrinogen:PCL fibers; 63 ± 29% and 120 ± 17%, respectively. They are statistically significantly different (unpaired *t*-test, *p* < 0.0001 (****), *N* = 47). (**C**,**D**) Histograms showing the maximum strain of blended fibrinogen:PCL nanofibers; (**C**) 75:25 (*N* = 47); (**D**) 25:75 (*N* = 47). (**E**,**F**) Maximum stress (breaking stress) of blended fibrinogen:PCL fibers as function of fiber diameter. The data were fit with a power law function $\sigma_{max}\left(D\right)=a\cdot D^{-b}+c$, where $\sigma_{max}$ is the maximum stress, *D* is the fiber diameter and *a*, *b*, *c* are fitting parameters.

**Figure 3 nanomaterials-13-01359-f003:**
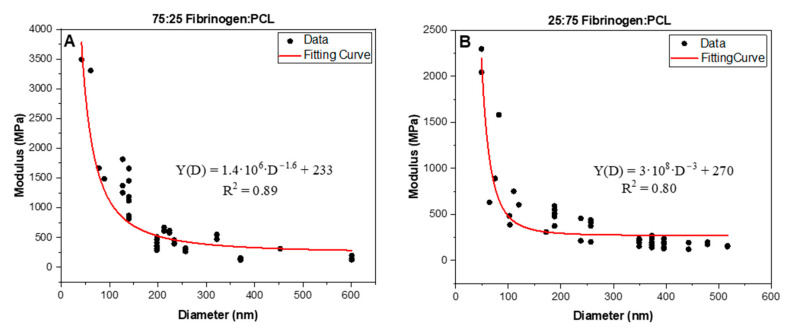
Young’s modulus as a function of fiber diameter of electrospun fibrinogen:PCL nanofibers with two different ratios (**A**) 75:25 (*N* = 43), (**B**) 25:75 (*N* = 44), *N* is the number of data points. The Young’s modulus increases drastically as the diameter decreases below 150 nm. However, the Young’s modulus only weakly depends on fiber diameter, *D*, for *D* > 150 nm. The graphs show the fit of a power law $Y\left(D\right)=a\cdot D^{-b}+c$, where *Y* is the modulus, *D* is the fiber diameter and *a*, *b*, *c* are fitting parameters.

**Figure 4 nanomaterials-13-01359-f004:**
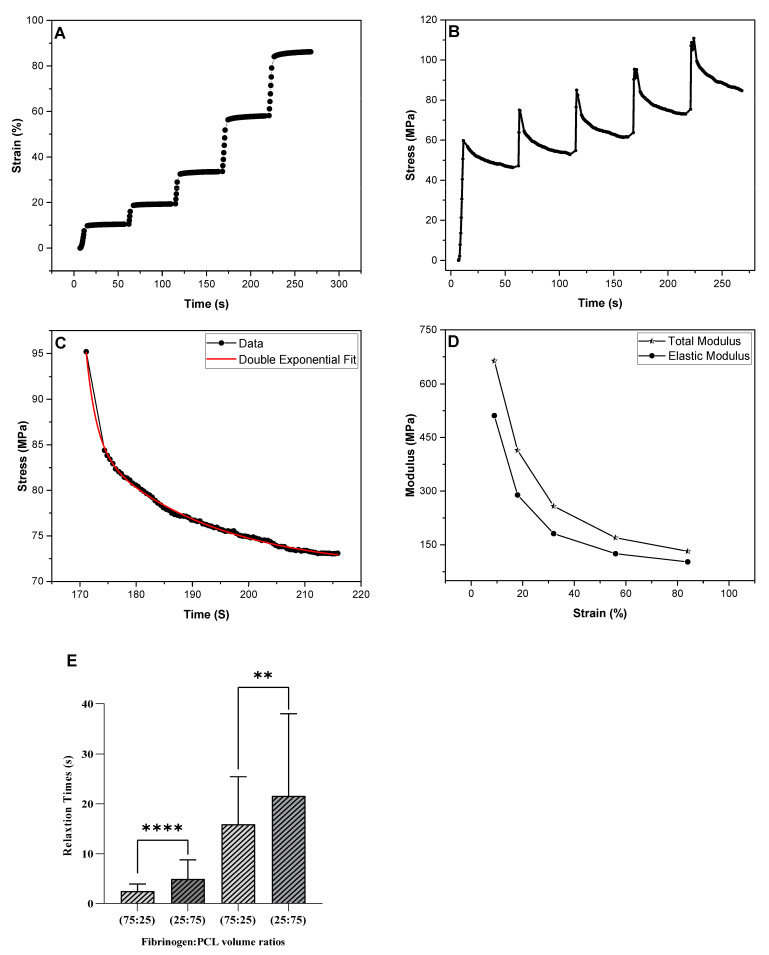
Incremental stress–strain curves. (**A**) Strain versus time curve for electrospun 75:25 fibrinogen:PCL fiber. The fiber was pulled to a small strain (~10%) and held constant for approximately 30–40 s; this process was repeated with a slightly larger strain at each time. (**B**) Stress versus time curve. At constant strain, the stress relaxes and decays exponentially with time. (**C**) Representative stress relaxation curves. A double exponential curve is fitted to the relaxation curve (R^2^ = 0.99) to determine the relaxation times. The fast and slow relaxation times for this curve were 1.8 s and 21 s. (**D**) Moduli versus strain curve. The total modulus, *Y_tot_*, (stars) and relaxed, elastic modulus, *Y*_0_, (dots) decrease as the strain increases. (**E**) The graph shows statistical differences between the slow and fast relaxation times of the fibers with two different ratios. The fiber diameter was 99 nm. ** indicates a *p*-value < 0.01; **** indicates a *p*-value < 0.0001.

**Figure 5 nanomaterials-13-01359-f005:**
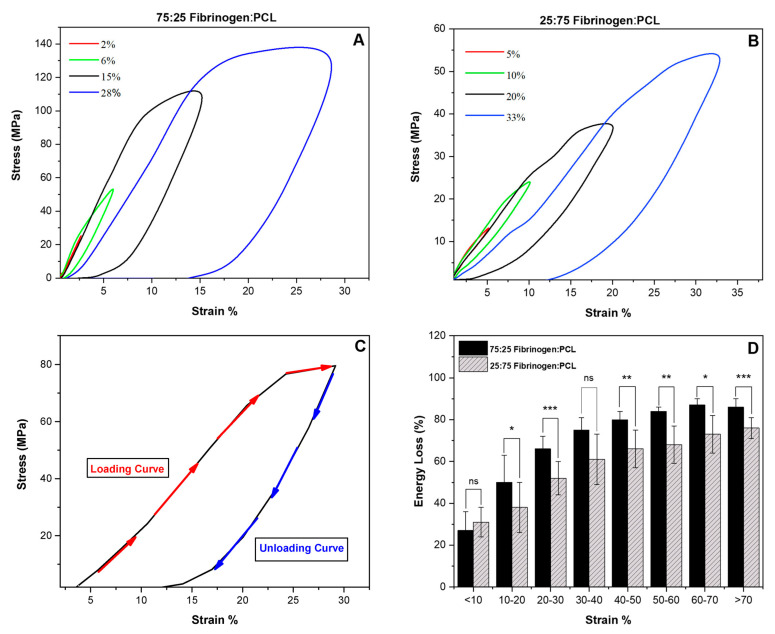
Stress and strain curves for four pulling cycles of fibrinogen:PCL fibers. (**A**) 75:25 ratio, (**B**) 25:75 ratio. The fiber was stretched to different strains until a permanent deformation occurred. The first and second pulls (red and green curves) the stress and strain returned to zero simultaneously. The third pull (black curve) represents the fiber prior to deformation, and the fourth pull (blue curve) represents the fiber after the permanent deformation. (**C**) Plot of a single stress–strain curve of electrospun fiber. The area enclosed by the loading and unloading curves is proportional to the amount of energy lost during the cyclical stretching process. (**D**) Energy vs. strain for fibers with different ratios. Error bars represent the standard deviation of the mean. * indicates a *p*-value < 0.05; ** indicates a *p*-value < 0.01; *** indicates a *p*-value < 0.001.

**Figure 6 nanomaterials-13-01359-f006:**
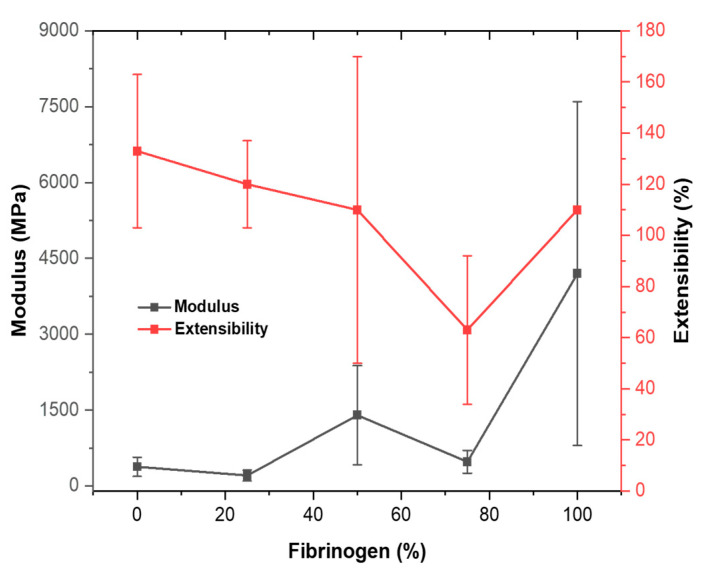
Extensibility and modulus of electrospun fibrinogen:PCL nanofibers at ratios of 0:100, 25:75, 50:50, 25:75 and 100:0.

**Table 1 nanomaterials-13-01359-t001:** Mechanical properties that exhibit ratio dependence, but no diameter dependence. Maximum strain (extensibility), $\varepsilon_{max}$, elastic limit, $\varepsilon_{elastic}$, and fast and slow relaxation times, $\tau_{1}$ and $\tau_{2}$ for electrospun 75:25 and 25:75 fibrinogen:PCL nanofibers.

Fgn:PCL (*V*/*V*)	$\varepsilon_{max}$ (%)	$\varepsilon_{elastic}$ (%)	$\tau_{1}$ (s)	$\tau_{2}$ (s)
(75:25)	63 ± 29	12 ± 4 to 27 ± 10	2.4 ± 1.4	14.2 ± 7.5
(25:75)	120 ± 17	18 ± 6 to 40 ± 14	4.7 ± 3.5	21.0 ± 15.9

**Table 2 nanomaterials-13-01359-t002:** Mechanical properties that exhibit diameter dependence, in addition to ratio dependence. Maximum stress, $\sigma_{max}$, Young’s modulus (initial slope of stress–strain curve), *Y*, total modulus $Y_{tot}$, and relaxed, elastic modulus, $Y_{0}$, for electrospun fibrinogen:PCL nanofibers with 75:25 and 25:75 ratios.

Fgn:PCL *V*/*V*	Diameter (nm)	*σ_max _* (MPa)	*Y * (MPa)	*Y_tot _* (MPa)	*Y_0 _* (MPa)
		$\sigma_{max}=5\cdot 10^{5}\cdot D^{-1.8}+30$	$Y=1.4\cdot 10^{6}\cdot D^{-1.6}+233$	$Y_{tot}=3\cdot 10^{6}\cdot D^{-1.8}+250$	$Y_{el}=5.6\cdot 10^{6}\cdot D^{-2}+250$
	50	5.0·10^2^	2.9·10^3^	2.9·10^3^	2.5·10^3^
75:25	100	1.7·10^2^	1.1·10^3^	1.0·10^3^	5.5·10^2^
	150	9.5·10^1^	6.9·10^2^	6.1·10^2^	4.5·10^2^
	200	6.9·10^1^	5.2·10^2^	4.6·10^2^	3.9·10^2^
		$\sigma_{max}=1.4\cdot 10^{6}\cdot D^{-2}+62$	$Y=3\cdot 10^{8}\cdot D^{-3}+2$70	$Y_{tot}=1\cdot 10^{8}\cdot D^{-2.7}+219$	$Y_{el}=5.7\cdot 10^{7}\cdot D^{-2.6}+200$
	50	6.2·10^2^	2.7·10^3^	2.8·10^3^	2.3. 10^3^
25:75	100	2.0·10^2^	5.7·10^2^	6.2·10^2^	5.6·10^2^
	150	1.2·10^2^	3.6·10^2^	3.5·10^2^	3.2·10^2^
	200	9.7·10^1^	3.1·10^2^	2.8·10^2^	2.6·10^2^

**Table 3 nanomaterials-13-01359-t003:** Mechanical properties of dry, electrospun fibrinogen:PCL nanofibers at different blending ratios. Maximum strain (extensibility), $\varepsilon_{max}$, maximum stress, $\sigma_{max}$, elastic limit, $\varepsilon_{el}$, Young’s modulus (initial slope of stress–strain curve), *Y*, relaxed, elastic modulus $Y_{0}$, and total modulus, $Y_{tot}$, fast and slow relaxation times, $\tau_{1}$ and $\tau_{2}$.

Fgn:PCL Ratio (*V*/*V*)	(100:0)	(75:25)	(50:50)	(25:75)	(0:100) *
$\varepsilon_{max}$(%)	110	63 ± 29	110 ± 60	120 ± 17	133 ± 30
$\sigma_{max}$(MPa)	2100 ± 1500	54 ± 19	410 ± 210	73 ± 34	348± 165
$\varepsilon_{el}$(%) *Y* (MPa) (Simple stress–strain)	15 ± 4.4 -- ***	12 ± 4 to 27 ± 10 372 ± 167	5 ± 5 1700 ± 800	18 ± 6 to 40 ± 14 274 ± 116	24 ± 10 265 ± 144
*Y*_0_ (MPa) (Incremental stress–strain)	3700 ± 3100	381 ± 189 **	980 ± 710	185 ± 95 **	317 ± 168 **
*Y_tot_* (MPa) (Incremental stress–strain)	4200 ± 3400	480 ± 225 **	1400 ± 980	208 ± 109 **	380 ± 192 **
$\tau_{1}$(s)	1.2 ± 0.4	2.4 ± 1.4	1.1 ± 0.4	4.7 ± 3.5	-- ****
$\tau_{2}$(s)	11 ± 5	14.2 ± 7.5	16 ± 6	21.0 ± 15.9	27 ± 17
References	[55]	This study	[56]	This study	[57]

* Information in the table is for PCL fibers with molecular weight of 114 kDa. In reference [57], the mechanical properties of electrospun PCL fibers with three different molecular weights were determined; only minor differences between the properties of the three different fibers were found. ** The maximum stress and the moduli values are for large diameter fibers, meaning diameters beyond which the strong diameter dependence levels off. For PCL fibers and 75:25 and 25:75 fibrinogen:PCL fibers, these are diameters larger than 150 nm. The moduli for fibers with a diameter below 150 nm showed a strong diameter dependence, whereas, above this value, the modulus showed a weak diameter dependence. *** Young’s modulus from simple stress–strain curves was not reported. It is estimated to be ~4000 MPa. **** Only one relaxation time was observed.

## Data Availability

The data presented in this study are openly available in Google drive, https://drive.google.com/drive/folders/19qFl9lv8WxscU2NcOHTJHy-mbtn0GgaK (accessed on 1 April 2023).

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
