# Peer review of "The Mechanical Properties of Blended Fibrinogen:Polycaprolactone (PCL) Nanofibers"

_nanomaterials, 2023, doi:10.3390/nano13081359_

Round 1

Reviewer 1 Report

Nouf Alharbi et al. fabricated a series of electrospun nanofibers from different blend ratios of fibrinogen and PCL, and further explored their mechanical properties. This manuscript is well organized and written. Some minor revisions should be conducted before publication.

1. The full name of PCL is suggested to be given in the title, in order to increase the readability.

2. Please state the reasons why fibrinogen was chosen in this study. What are the merits and advantages of fibrinogen compared with some other biopolymers like elastin, collagen, chitosan, etc.?

3. The merits of electrospinning technique should be further outlined, and some recent works about the innovative electrospinning like 10.3390/nano13071150 and 10.3390/nano13030593 published on Nanomaterials are suggested to be discussed.

4. How did the authors choose the parameters of electrospinning? Do they conduct any preliminary experiments?

Minor editing of English language is required.

Reviewer 2 Report

The manuscript titled “The Mechanical Properties of Blended Fibrinogen:PCL Nanofibers” by Alharbi, N.; et al. is an original research where the authors study mechanical properties of blended fibrinogen-polycaprolactone (PCL) fibers by single molecule techniques like atomic force microscopy coupled with an optical microscope. The tested conditions of this research (ratios of 25:75 and 75:25 of fibrinogen:PCL, respectively) are complementary with previous works and serve to have a more complete outlook of the mechanical performance of these materials according the aforementioned fibrinogen:PCL ratio and the fiber diameter. The most relevant outcomes gathered in the present work could have a positive impact on many industrial sectors, such as drug-delivery, textile or solar cells, among others. The achieved results are well-discussed during the main body of the reported manuscript. The scientific paper is well written. In my opinion the present manuscript is innovative and the methodological approached used matches with the scope of Nanomaterials. For the above described reasons, I will recommend the publication in Nanomaterials once the following remarks are fixed:

--------

INTRODUCTION

Introduction section is clear and concise. Some minor remarks must be addressed in order to increase the quality of the scientific content shown in the present manuscript:

“Large surface area to volume ratios, (…) make electrospun fibers an attractive material for various fields (…) and filtration” (lines 28-31). Here, the authors should also indicate other alternative promising uses of electrospun fibers as batteries, fuel cells or solar cells [1].

[1] Sun, G.; et al. Electrospinning of Nanofibers for Energy Applications. Nanomaterials 20166, 129. https://doi.org/10.3390/nano6070129.

“Studies showed that blending natural (…) than synthetic ones” (lines 46-48). Please, the authors should provide a brief feasible explanation for this statement like the fact of interlinkage bonding between the natural and synthetic polymers which render and improvement of the mechanical performance.

“Moreover, electrospun PCL networks can mimic the structure of the native extracellular matrices (ECM)” (lines 61-62). Here, it lacks a relevant reference of some previous recent research devoted in this field [2]

[2] Unal, S.; et al. Polycaprolactone/Gelatin/Hyaluronic Acid Electrospun Scaffolds to Mimic Glioblastoma Extracellular Matrix. Materials 202013, 2661. https://doi.org/10.3390/ma13112661.

“The ideal scaffold should mimic (…) extracellular matrix (ECM)” (lines 66-67). Please, the authors have already defined the abbreviation extracellular matrix above in the line 62.

“One of the most suitable tools to measure the mechanical properties (…) atomic force microscope (AFM)” (lines 80-81). Here, the authors should describe that AFM is capable to determine the mechanical properties of soft matter samples [3] through the analysis of the recorded force spectroscopy curves [4] of the indentated sample surface areas.

[3] Magazzù, A.; et al. Investigation of Soft Matter Nanomechanics by Atomic Force Micrsocopy and Optical Tweezers: A Comprehensive Review. Nanomaterials 202313, 693. https://doi.org/10.3390/nano13060963.

[4] Lostao, A.; et al. Recent advances in sensing the inter-biomolecular interactions at the nanoscale – A comprehensive review of AFM-based force spectroscopy. Int. J. Biol. Macromol2023238, 124089. https://doi.org/10.1016/j.ijbiomac.2023.124089.

--------

MATERIALS AND METHODS

Materials and methods employed by the authors are unequivocally described which is crucial to mimic the same experimental approach in other labs placed in different locations. Only the following remarks should be fixed:

“atomic force microscopy (AFM) probe” (line 132-133). What are the characteristics of the AFM probe used by the authors in this work (nominal tip radius)? Other relevant information was already specified by the authors in the lines 162-163.

“using an accelerating votage of 5 kV” (lines 292-293). Please, the authors should modify “5 kV” by “5 keV”.

“The topography images, (…) Gwyddion software program” (lines 297-298). Here, the authors should cite the following reference [5].

[5] Neças, D.; et al. Gwyddion: an open-source software for SPM data analysis. Cent. Eur. J. Phys201210, 181-188. https://doi.org/10.2478/s11534-011-0096-2.

“The fibers had an average diameter, (…), respectively” (lines 299-300). (OPTIONAL), this statement could be shifted to the corresponding Results section.

--------

RESULTS

Authors perfectly state the most relevant outcomes found in the present work. Nevertheless, there exists some remarks that the authors should fix out.

I)        Figure 2, panels E and F (line 329). Could the authors add the regression coefficient (R2) to better visualize the good agreement between the experimental and fitted data? Same comment for Fig. 3 (line 397). This information is already available in the main manuscript body text but if it also appears in the figure could aid to the potential readers.

II)     “(…) to be 2 ± 1 s and 14.2 ± 7.5 s (…) and 4.7 ± 3.5 s and 21 ± 16 s (…)” (lines 439-440). Please, the authors should homogenize the significant figures. This comment should be taken into account for the rest of the main manuscript body text.

III)  Figure 5, panel D (line 527). Did the authors carry out statistical analysis of the energy-strain data recorded for the 75:25/25:75 fibrinogen:PCL samples? This information should be provided in order to ascertain if the observed differences are statistically significant.

--------

DISCUSSION

“Table 3 summaries key mechanical properties (…)” (line 573). Please, the authors should modify this statement by “Table 3 summarizes the key mechanical properties (…)”.

Figure 6 (line 591). Please, the authors should add the respective standard deviation (SD) bars.

--------

CONCLUSIONS

The authors perfectly states this section. No actions are requested.

(OPTIONAL) Nevertheless, it may be interesting if the authors point out some future avenues to pursue this research.

--------

REFERENCES

Bibliography citations are not in the proper format of Nanomaterials. The journal name should appear in abbreviated form and Italics. The publication year and journal volume should be highlighted in bold and Italics, respectively. Additionally, the journal issue should be erased. The authors should take care of these points.

--------

OVERVIEW AND FINAL COMMENTS

The submitted work is well-designed and the gathered results are interesting to better understand the impact of the electrospun fibrinogen:polycaprolactone nanofiber diameter at different ratios of fibrinogen and PCL on the mechanical properties. This knowledge will aid to design more durable materials which can be used for many Industrial applications. For these reasons, I will recommend the present scientific manuscript for further publication in Nanomaterials once all the aforementioned suggestions will be properly fixed.

The scientific paper is well-written. No extensive language modifications are required. 
